# Does peer education go beyond giving reproductive health information? Cohort study in Bulawayo and Mount Darwin, Zimbabwe

Aveneni Mangombe ![ORCID] ,[1] Philip Owiti,[2,3] Bernard Madzima,[1] Sinokuthemba Xaba,[4] Talent M Makoni ![ORCID] ,[4] Kudakwashe C Takarinda ![ORCID] ,[4,5] Collins Timire,[6] Anesu Chimwaza,[4] Mbazi Senkoro,[7] Simbarashe Mabaya,[8] Julia Samuelson,[9] Wole Ameyan,[9] Talent Tapera,[10] Nonhlanhla Zwangobani,[11] Jaya Prasad Tripathy,[12] Ajay M V Kumar[2,13]

For numbered affiliations see end of article.

**Correspondence to**
Aveneni Mangombe;
mangombeaveh@gmail.com

## ABSTRACT

**Objective** Peer education is an intervention within the voluntary medical male circumcision (VMMC)–adolescent sexual reproductive health (ASRH) linkages project in Bulawayo and Mount Darwin, Zimbabwe since 2016. Little is known if results extend beyond increasing knowledge. We therefore assessed the extent of and factors affecting referral by peer educators and receipt of HIV testing services (HTS), contraception, management of sexually transmitted infections (STIs) and VMMC services by young people (10–24 years) counselled.

**Design** A cohort study involving all young people counselled by 95 peer educators during October–December 2018, through secondary analysis of routinely collected data.

**Setting** All ASRH and VMMC sites in Mt Darwin and Bulawayo.

**Participants** All young people counselled by 95 peer educators.

**Outcome measures** Censor date for assessing receipt of services was 31 January 2019. Factors (clients' age, gender, marital and schooling status, counselling type, location, and peer educators' age and gender) affecting non-referral and non-receipt of services (dependent variables) were assessed by log-binomial regression. Adjusted relative risks (aRRs) were calculated.

**Results** Of the 3370 counselled (66% men), 65% were referred for at least one service. 58% of men were referred for VMMC. Other services had 5%–13% referrals. Non-referral for HTS decreased with clients' age (aRR: ~0.9) but was higher among group-counselled (aRR: 1.16). Counselling by men (aRR: 0.77) and rural location (aRR: 0.61) reduced risks of non-referral for VMMC, while age increased it (aRR ≥1.59). Receipt of services was high (64%–80%) except for STI referrals (39%). Group counselling and rural location (aRR: ~0.52) and male peer educators (aRR: 0.76) reduced the risk of non-receipt of VMMC. Rural location increased the risk of non-receipt of contraception (aRR: 3.18) while marriage reduced it (aRR: 0.20).

**Conclusion** We found varying levels of referral ranging from 5.1% (STIs) to 58.3% (VMMC) but high levels of

## Strengths and limitations of this study

► This study included all the clients who were counselled by peer educators during the study period in the two project districts.
► It used routine programme data, thus making the findings a likely true reflection of the situation.
► The conduct and reporting of the study adhered to Strengthening the Reporting of Observational Studies in Epidemiology guidelines.
► The study could not exclude clients not in need of referral from the analysis due to lack of data from the peer educator registers in determining the reason for non-referral.
► Due to the small sample size of those referred for sexually transmitted infection diagnosis and treatment, the study lacked statistical power to carry out further analyses.

receipt of services. Type of counselling, peer educators' gender and location affected receipt of services. We recommend qualitative approaches to further understand reasons for non-referrals and non-receipt of services.

## INTRODUCTION

HIV still remains a major global public health concern, with 1.8 million new infections and 940 000 deaths reported in 2017.[1] Of concern is the slower decline of AIDS-related illnesses and deaths among young people (10–24 years) compared with adults.[2] Coverage of HIV testing and access to treatment remain significantly low among young people, especially in sub-Saharan Africa.[2] Studies and systematic reviews have reported high rates of sexually transmitted infections (STIs), teenage pregnancy and suicide attempts, coupled with poor receipt

of sexual and reproductive health (SRH) and HIV services among young people.[2 3]

Zimbabwe is a low-income country with a population of nearly 13.5 million as of 2017.[4] A third of the country's population are young people aged 10–24 years.[5] Mirroring the global scenario, the Zimbabwe Demographic Health Survey 2015 reported a high fertility rate among young girls (15–19 years), low coverage of HIV testing among adolescents aged 15–19 years (35%–46%) and low comprehensive knowledge on HIV.[6] Though SRH and HIV interventions focused on young people have been in existence for the past three decades, in Zimbabwe, the first 5-year strategic plan was developed in 2010. Among other things, this phase defined a minimum package of interventions that recognised the role of community-based youth peer educators in educating young people on SRH and HIV.

There is no universally agreed upon definition of peer education. We define it as a structured process of sharing of relevant information, values and behaviours among members of similar status, in an appropriate setting for both the educator and the learner. Following a series of reviews, peer education has been noted to be an effective youth-led approach for influencing positive behavioural outcomes among beneficiaries, if given appropriate support systems and contextualised to different settings and needs of beneficiaries.[7 8] Peer education has been established to be more effective in reaching out to key populations (eg, adolescents selling sex and who are sexually exploited) and delivering messages that are considered taboo to be delivered through schools, religious and family settings.[9] Integration of peer education interventions with holistic and well-coordinated interventions makes it more effective towards improving health outcomes among beneficiaries in different contexts.[10 11]

In 2016, a project to link voluntary medical male circumcision (VMMC) with the adolescent sexual and reproductive health (ASRH) services (Smart LyncAges Project) was started in Bulawayo city and Mount Darwin district. With a special focus on VMMC, the project targets both men and women for ASRH services. In 2017, the scope of the project was expanded to engage youth peer educators in promoting referral and receipt of SRH and VMMC services, informed by Michielsen *et al*'s review.[12]

Since the redesign, no study has been conducted to determine the effect of the new peer education model on the uptake of SRH and VMMC services, beyond just providing information to their peers. Within this background, we carried out a study to assess the referral and receipt of HIV testing services (HTS), contraception, diagnosis and treatment of STIs and VMMC services among young people (10–24 years) counselled by peer educators and their associated factors under the Smart LyncAges Project.

## METHODS

### Study design and study population

This was a cohort study involving young people aged 10–24 years (both men and women) counselled by 95 peer educators under the VMMC–ASRH linkages project in Bulawayo city and Mount Darwin district of Zimbabwe during October–December 2018. Secondary data routinely collected under the project on peer education and clients' service uptake were analysed.

### Setting

#### General setting

In 2009, Zimbabwe devised the National ASRH strategy 2010–2015 to promote adoption of safer SRH practices and to increase availability, access and use of SRH and HIV services by young people.[13] The strategy outlined three settings for providing 'friendly' SRH and HIV services: health facility, community and school-based. The health facility approach required every facility to establish and equip special rooms (youth-friendly corners). The community approach involved establishment of 'community youth centres' (CYCs), while the school-based approach focused on the provision of life skills education and counselling mainly by teachers.

In 2015, an extensive review of the 2010–2015 interventions in Zimbabwe was conducted so as to inform the development of the National ASRH Strategy II: 2016–2020.[14] One of the review's conclusions acknowledged peer education as an important tool in ASRH programming, though with some modifications.[15] This strategy seeks to strengthen comprehensive sexuality education (CSE) and provision of quality-assured and adolescent-friendly services, delivered through schools and colleges, public health facilities and the community. As part of expanding CSE in both the in-school and out-of-school settings, Zimbabwe aligned its tools and training materials with the 2018 UNESCO revised international technical guidance on sexuality education.[16] The minimum package of SRH services for young people includes contraception, STI diagnosis and treatment, HTS and integration of VMMC.

#### VMMC–ASRH linkages project

The VMMC–ASRH linkages project has been implemented in Bulawayo and Mount Darwin, Zimbabwe, since 2016. Bulawayo is an urban city with 27 public health facilities, while Mount Darwin is a rural district with 19 public health facilities. In Mount Darwin, VMMC services are provided at the Mount Darwin Hospital, while diagnosis and treatment of STIs, contraception and HTS are being provided at Mount Darwin Hospital and two CYCs (Mount Darwin and Dotito) supported through the Zimbabwe National Family Planning Council. In Bulawayo, VMMC services are provided at the Bulawayo Male Circumcision (MC) site and the Lobengula MC site, and sometimes through outreach camps in the 15 CYCs and clinics. The CYCs in Bulawayo are primarily focused on imparting information and counselling services, edutainment services through films, drama and sports, library

services, and vocational and life skills training. All the service delivery points under this project were oriented on the VMMC–ASRH linkages service delivery protocols and ASRH. The VMMC–ASRH service delivery protocols clearly highlight the scope of work, reporting, referral, coordination and supervision mechanisms for peer educators. Demand creation for both SRH and VMMC services for young people relies heavily on community-based peer educators and VMMC mobilisers. Though the project has a special focus on men for VMMC, it targets both men and women for ASRH services.

In 2017, the Smart LyncAges Project was redesigned and expanded the work of youth peer educators in promoting referral and receipt of SRH and VMMC services. The project also provided an updated training that introduced relevant tools for delivering the messages (distribution of information, education and communication materials, and social media platforms such as Facebook and WhatsApp), a clearly defined referral pathway (including referral forms, tracking and two-way feedback mechanism) and provision of adequate tools for documentation.

## Peer education

Each CYC is expected to coordinate and supervise the work of five to six peer educators at any point in time. Peer educators are male and female volunteers aged 10–24 years, residing in the community and nominated by the young people in stakeholder community meetings. They must be able to read and write in English and the respective local language and should have passed at least three subjects at the end of secondary level (13th grade). Under the Smart LyncAges Project, peer educators undergo a 7-day standard training, which also addresses the referral pathway. Each peer educator is attached to the nearest CYC and allocated a catchment area to cover. Peer educators are expected to contribute at least 2 hours a day for at least 3 days a week to the project. Active peer educators receive a fixed monthly allowance of US$15, paid on submission of daily and monthly summary reports of their activities.

While peer educators usually spend most of their time in the community conducting outreach sessions in pairs, they do sit in CYCs on a rotation basis to cater to the walk-in clients. At CYCs, considerations are made to ensure that at least one male and one female peer educator is available during the operating hours. Peer educators reach clients through both individual (*one-on-one*) and group counselling sessions (either individually or in pairs). The average duration of group counselling sessions is 1 hour, while the size ranges from 15 to 25 participants. Depending on the sensitivity of the subject matter/topic, young people are usually grouped into clusters of 10–14, 15–19 and 20–24 years. Both peer educators and young people agree on the choice of places for group counselling sessions, such as recreation parks, playgrounds, schools and community halls where young people usually congregate. In addition to providing information and counselling services,

peer educators also distribute condoms to the clients. However, they refer clients for such when they run out of stock or are in settings (such as churches and schools) where condom distribution is prohibited.

Peer educators also facilitate referrals to ASRH and VMMC sites. The referral process includes providing information regarding location of the service delivery point, hours of operation, user fees (if any) and details of the contact person (if available). They also complete referral forms in triplicate (one copy kept for records and two copies sent with the client to the destination service delivery point). One copy of the latter is retained at the service delivery point, while the other is given to the client, who is expected to return it to the respective peer educator. The form retained at the service delivery point is sent to the respective CYC by post or hand-delivered. This system facilitates feedback on receipt of services by referred clients. Monthly, the peer educators are expected to track all the referred clients, through verification of redeemed referral forms, physical home visits or by telephone (where possible). In some cases, peer educators accompany clients to the service delivery points to ensure that they avail the service.

As part of documentation, each peer educator maintains individualised 'daily and monthly summary peer educator registers', where they document the sociodemographic details of the clients who underwent counselling, the referrals and receipt of the services. Each service delivery point under the VMMC–ASRH linkages project also uses primary registers for SRH and HIV services provision.

## Data variables, sources of data and data collection

A structured data collection tool was used to collect the following data variables: sociodemographic characteristics of clients (age, sex, marital status, location, schooling status and type of counselling sessions undergone), the age and sex of their peer educators, whether referred or not, services referred for and if the clients received the services. The source of data was the peer educator registers. 'Referral' in this study meant that clients consented to receive some postcounselling services (such as HIV testing, VMMC, contraception or STI diagnosis and treatment) and were given a referral form. 'Receipt' of a service meant receiving/getting the services at a referral service delivery point by the young person who had been referred by peer educators (between October and December 2018) by end of January 2019.

## Data analysis and statistics

Data entry and validation was performed using EpiData software V.4.4.1.0 (EpiData Association, Odense, Denmark),while analyses were carried out using EpiData Analysis V.2.2.2.186 and STATA V.14 software. Proportions were used to summarise referrals and receipt of services for those referred. There were four major services for which the clients were referred: VMMC (for men only), HTS, STI diagnosis and treatment, and contraception

(for both men and women). Thus, the referral and receipt of these services were the key outcome variables for this study. Of these, factors associated with non-referral were assessed for HTS and VMMC services only. Factors associated with non-referrals for contraception and STI diagnosis and treatment were not analysed as it was not possible to establish the appropriate denominator defining the eligibility for these services. Factors associated with non-receipt of services were assessed for VMMC and contraceptive use only. Non-receipt of HTS was not analysed as HTS can also be provided as an opt-out provider-initiated testing and counselling service at all the service delivery points, irrespective of the reason for which the client was referred. HTS was also a necessary service prior to conducting VMMC. Non-receipt of STI services among those referred was not analysed due to small the sample size. The strength of associations was initially expressed using unadjusted relative risks (aRRs) and then further expressed using aRRs and 95% CIs, using log-binomial regression methods. Two-sided $p<0.05$ were considered statistically significant.

## RESULTS
There were 95 peer educators (52% men) in the study sites with a median age of 22 years (range: 15–24). A total of 3370 young people (2207 men and 1163 women) received counselling services from the peer educators (table 1). Forty per cent of young people were aged 15–19 years, with the majority being male (66%) and single (98%). Majority were still in school (69%) and underwent group counselling (78%).

### Referrals and receipt of services
Table 2 shows the proportions of clients who were referred and those who received the services among those referred. Of the 3370 counselled young people, sixty five per cent (75% of men and 47% of women) were referred for SRH services. Of those referred, 77% had been referred for only one service, with the rest being referred for two or more services. Seventy five per cent (75% of men and 76% of women) of those referred received the services they had been referred for.

Of the men counselled, 58% were referred for VMMC services, of whom 69% received the services. The other services for which the adolescents were referred for included HIV testing (13%), contraception (13%), and diagnosis and treatment of STIs (5%). Among the services referred for, the receipt of STI services was lowest (39%).

### Factors associated with non-referrals for HTS and VMMC
#### Non-referral for HTS
In multivariable analysis, only age of client and type of counselling were significantly associated with non-referral for HTS, holding other factors that influence HTS referral and receipt constant. Adolescents aged 15–19 years had a 9% (95% CI 0.81% to 1.03%) reduced risk of non-referral for HTS as compared with those aged

| Table 1 | Sociodemographic characteristics of young people counselled by peer educators in Bulawayo and Mount Darwin, Zimbabwe (October–December 2018) |
|---|---|
| **Characteristics** | **n (%)** |
| Total | 3370 (100) |
| Age of young people (years) | |
| 10–14 | 1242 (36.9) |
| 15–19 | 1346 (39.9) |
| 20–24 | 782 (23.2) |
| Sex of young people | |
| Male | 2207 (65.5) |
| Female | 1163 (34.5) |
| Marital status of young people | |
| Single | 3288 (97.6) |
| Married | 77 (2.3) |
| Divorced/separated | 5 (0.1) |
| Schooling status of young people | |
| In-school | 2334 (69.3) |
| Out of school | 1036 (30.7) |
| Type of counselling session received | |
| Individual | 741 (22.0) |
| Group | 2629 (78.0) |
| Type of setting | |
| Bulawayo (urban) | 2160 (64.1) |
| Mount Darwin (rural) | 1210 (35.9) |
| Age of peer educator (years)* | |
| 15–19 | 751 (22.3) |
| 20–24 | 2619 (77.7) |
| Sex of peer educators | |
| Male | 49 (51.6) |
| Female | 46 (48.4) |
| Sex of peer educators reach† | |
| Male | 1626 (48.2) |
| Female | 1744 (51.8) |

*Refers to the number of clients who were counselled by the peer educators aged 15–19 and 20–24 years.
†Refers to the number of clients who were counselled by male and female peer educators.

10–14 years old. Those who underwent group counselling had 16% (95% CI 1.04% to 1.27%) increased risk of non-referral for HTS. Sex and marital status of the peer educators did not influence referral for HTS (table 3).

### Non-referral for VMMC services
Compared with clients in the age group of 10–14 years, those in the age groups of 15–19 and 20–24 years had 59% (95% CI 1.34% to 1.87%) and 83% (95% CI 1.49% to 2.26%) increased risk of non-referral for VMMC, respectively. Clients in the rural district of Mount Darwin were at 39% (95% CI 0.52% to 0.71%) reduced risk of

Table 2  Services for which counselled young people were referred for by peer educators and received in Bulawayo and Mount Darwin districts, Zimbabwe (October–December 2018)

| Type of service | | Referred* | | Received* | |
|---|---|---|---|---|---|
| | | n | (%)† | n | (%)‡ |
| Referred for any service | | 2191 | (65.0) | 1645 | (75.1) |
| | Females | 544 | (46.8) | 413 | (75.9) |
| | Males | 1647 | (74.6) | 1232 | (74.8) |
| HTS | | 424 | (12.6) | 271 | (63.9) |
| | Females | 154 | (13.2) | 88 | (57.1) |
| | Males | 270 | (12.2) | 183 | (67.8) |
| VMMC (among males only, n=2207) | | 1287 | (58.3) | 881 | (68.5) |
| Contraception§ | | 452 | (13.4) | 363 | (80.3) |
| | Females | 206 | (17.7) | 161 | (78.2) |
| | Males | 246 | (11.2) | 202 | (82.1) |
| STI diagnosis and treatment | | 171 | (5.1) | 67 | (39.2) |
| | Females | 87 | (7.5) | 28 | (32.2) |
| | Males | 84 | (3.8) | 39 | (46.4) |
| Other SRH and HIV service | | 497 | (14.7) | 397 | (79.9) |
| | Females | 251 | (21.6) | 207 | (82.5) |
| | Males | 246 | (11.2) | 190 | (77.2) |

*Percentages may not add up to 100% for some clients were referred for more than one service.
†Denominator is the total number counselled (n=3370), except for VMMC.
‡Denominator is the total number referred in the respective category.
§Contraception refers to condoms, oral pills, injectables and implants.
HTS, HIV testing services; SRH, sexual and reproductive health; STI, sexually transmitted infection; VMMC, voluntary male medical circumcision.

non-referral compared with those in the urban city of Bulawayo. Clients counselled on VMMC by male peer educators were at 23% (95% CI 0.67% to 0.80%) reduced risk of non-referral as compared with those counselled by female peer educators (table 4).

### Factors associated with non-receipt for VMMC and contraception

#### Non-receipt of VMMC services

Clients referred by male peer educators had a 24% (95% CI 0.62% to 0.94%) lower risk of non-receipt of services than those referred by their female counterparts. Those referred from the rural district of Mount Darwin (aRR: 0.52, 95% CI 0.41 to 0.67) and those referred through group counselling sessions (aRR: 0.52, 95% CI 0.41 to 0.66) had a significantly lower risk of non-receipt of VMMC services as compared with urban Bulawayo and individual counselling sessions, respectively. The age of the clients did not influence receipt of VMMC services (table 5).

#### Non-receipt of contraception services

Clients referred from the rural district had 3.2 times (95% CI 1.93 to 5.22) higher risk of non-receipt of contraception services as compared with the urban-based counterparts. Married clients had 80% (95% CI 0.07% to 0.58%) reduced risk of not receiving contraception services than the single clients (table 6).

### DISCUSSION

This study, the first one to assess the role of peer educators in the referral and receipt of selected SRH and VMMC services among young people in Zimbabwe, found relatively moderate levels of referrals, particularly for VMMC services. A higher proportion of women/girls were referred for each of the individual services as compared with men/boys. However, overall, more men were referred for any service compared with women, likely due to VMMC (for men only). However, for those referred, receipt of the services was high except for STI diagnosis and treatment. Majority of the referred clients had been referred for just one service, perhaps indicating limited counselling or reflecting the lack of need. The risk of not being referred for HTS decreased slightly with older age of clients, though for VMMC services, younger aged adolescents were more likely to be referred. Slightly increased non-referral for HTS was observed in clients who underwent group counselling sessions as compared with individual sessions.

As regards receipt of services, generally more men received the services referred for as compared with women. With regard to VMMC services, those counselled by male peer educators were more likely to be referred and to receive, likewise to the clients in the rural district. Rural setting increased the risk of not receiving contraception services, while being married reduced this risk.

The low referral for diagnosis and treatment of STIs might have been influenced by (1) the lack of confidence of peer educators in determining the need for referral and (2) limited self-reports of STI-like symptoms. In any population, only a minority will have STI or STI-like symptoms and referrals rates may be low. However, the low receipt of the diagnosis and treatment of STI services among the referred clients might have been due to the associated user fees levied (mostly in council clinics and hospitals) in both rural and urban settings.[17 18] Abolishing or largely subsidising these costs while maintaining or sustaining supply and diagnostics may improve receipt of services by young people.

Low-risk perception among early adolescents has been established to be one of the barriers for referral and receipt of HTS.[19] Other studies have also shown that demand for HTS is reduced in adolescents who require parental/guardian consent, especially due to the perceived negative reactions from parents/guardians.[19] Zimbabwe's age of consent for HTS at 16 years may explain the reduced non-referral for HTS as the client's age increases. Individualised or client-centred counselling has been established

**Table 3** Factors associated with non-referral for HTS among counselled young people in Bulawayo and Mount Darwin, Zimbabwe (October–December 2018)

| Sociodemographic characteristics | Total counselled | Not referred for HTS | | Unadjusted | | Adjusted | |
|---|---|---|---|---|---|---|---|
| | | n | (%) | RR | (95% CI) | aRR | (95% CI) |
| Total | 3370 | 2946 | (100) | | | | |
| Age (years) | | | | | | | |
| 10–14 | 1242 | 1178 | (94.8) | Ref | | Ref | |
| 15–19 | 1346 | 1134 | (84.2) | **0.89** | **(0.82–0.96)** | **0.91** | **(0.83–0.99)** |
| 20–24 | 782 | 634 | (81.1) | **0.85** | **(0.78–0.94)** | 0.92 | (0.81–1.03) |
| Sex | | | | | | | |
| Male | 2207 | 1937 | (87.8) | 1.01 | (0.93–1.01) | 0.98 | (0.90–1.05) |
| Female | 1163 | 1009 | (86.8) | Ref | | Ref | |
| Marital status | | | | | | | |
| Single | 3288 | 2889 | (87.9) | Ref | | Ref | |
| Married | 77 | 52 | (67.5) | 0.77 | (0.58–1.01) | 0.88 | (0.67–1.17) |
| Divorced or separated | 5 | 5 | (100.0) | 1.14 | (0.47–2.74) | 1.35 | (0.58–3.25) |
| School status | | | | | | | |
| In-school | 2334 | 2118 | (90.7) | Ref | | Ref | |
| Out of school | 1036 | 828 | (79.9) | **0.88** | **(0.81–0.95)** | 0.95 | (0.86–1.05) |
| Counselling session | | | | | | | |
| Individual | 741 | 554 | (74.8) | Ref | | Ref | |
| Group | 2629 | 2392 | (91.0) | **1.22** | **(1.11–1.33)** | **1.16** | **(1.04–1.27)** |
| Type of setting | | | | | | | |
| Bulawayo (urban) | 2160 | 1958 | (90.6) | Ref | | Ref | |
| Mount Darwin (rural) | 1210 | 988 | (81.7) | **0.90** | **(0.83–0.97)** | 0.93 | (0.85–1.01) |
| Age of peer educator (years)* | | | | | | | |
| 15–19 | 751 | 673 | (89.6) | Ref | | Ref | |
| 20–24 | 2619 | 2273 | (86.8) | 0.97 | (0.89–1.05) | 0.96 | (0.85–1.03) |
| Sex of peer educator† | | | | | | | |
| Male | 1624 | 1421 | (87.5) | 1.00 | (0.93–1.07) | 1.01 | (0.93–1.08) |
| Female | 1746 | 1525 | (87.3) | Ref | | Ref | |

In bold: statistically significant at p<0.05.
*Refers to the number of clients who were counselled by peer educators aged 15–19 and 20–24 years.
†Refers to the number of clients who were counselled by male and female peer educators.
aRR, adjusted relative risk; HTS, HIV testing services; Ref, reference; RR, relative risk.

to be a more effective approach for HTS and post-test services than group counselling,[20] as demonstrated also by this study.

Group counselling sessions may have provided an opportunity for client peer influence on receipt of VMMC services. Counselling by male peer educators may also encourage openness among male clients, leading to more referral and receipt of VMMC services. As regards contraception, urban areas present more convenient service delivery points for accessing contraceptives than rural areas, explaining why non-receipt is lower in the urban setups. Myths on the association between modern contraception and future infertility[21 22] may have resulted in the higher non-receipt rates of contraception among the adolescent girls referred. This belief may be more common in the rural setups and at times is propagated by other health workers.[17 22]

**Strengths**
This study had several strengths. We included all the clients who were counselled by peer educators during the study period in the two project districts, which were the only districts implementing this intervention in the whole country. As we used routine programme data, the findings are likely a true reflection of the situation. The data were extracted into standard proformas by the principal investigator and trained data collectors; this enhanced quality. Lastly, the conduct and reporting of the study adhered to Strengthening the Reporting of Observational Studies in Epidemiology cohort reporting guidelines.[23]

**Table 4** Factors associated with non-referral for VMMC services among counselled young people in Bulawayo and Mount Darwin, Zimbabwe (October–December 2018)

| Sociodemographic characteristics | Total counselled | Not referred for VMMC | | Unadjusted | | Adjusted | |
|---|---|---|---|---|---|---|---|
| | | n | (%) | RR | (95% CI) | aRR | (95% CI) |
| Total | 2207 | 920 | (100) | | | | |
| **Age (years)** | | | | | | | |
| 10–14 | 944 | 287 | (30.4) | Ref | | Ref | |
| 15–19 | 795 | 362 | (45.5) | **1.50** | **(1.28–1.74)** | **1.59** | **(1.34–1.87)** |
| 20–24 | 468 | 271 | (57.9) | **1.90** | **(1.61–2.25)** | **1.83** | **(1.49–2.26)** |
| **Marital status** | | | | | | | |
| Single | 2179 | 904 | (41.5) | Ref | | Ref | |
| Married | 27 | 15 | (55.6) | 1.34 | (0.80–2.23) | 1.31 | (0.78–2.22) |
| Divorced or separated | 1 | 1 | (100) | 2.41 | (0.34–17.1) | 2.46 | (0.34–1.77) |
| **School status** | | | | | | | |
| In-school | 1613 | 604 | (37.4) | Ref | | Ref | |
| Out of school | 594 | 316 | (53.2) | **1.42** | **(1.24–1.63)** | 1.09 | (0.92–1.30) |
| **Counselling session** | | | | | | | |
| Individual | 427 | 206 | (48.2) | Ref | | Ref | |
| Group | 1780 | 714 | (40.1) | **0.83** | **(0.71–0.97)** | 0.86 | (0.73–1.01) |
| **Type of setting** | | | | | | | |
| Bulawayo (urban) | 1392 | 644 | (46.3) | Ref | | Ref | |
| Mount Darwin (rural) | 815 | 276 | (33.9) | **0.73** | **(0.64–0.84)** | **0.61** | **(0.52–0.71)** |
| **Age of peer educator (years)\*** | | | | | | | |
| 15–19 | 507 | 217 | (42.8) | Ref | | Ref | |
| 20–24 | 1700 | 703 | (41.4) | 0.97 | (0.83–1.13) | 0.92 | (0.78–1.08) |
| **Sex of peer educator†** | | | | | | | |
| Male | 1159 | 440 | (38.0) | **0.83** | **(0.73–0.94)** | **0.77** | **(0.67–0.80)** |
| Female | 1048 | 480 | (45.8) | Ref | | Ref | |

In bold: statistically significant at p<0.05.

\*Refers to the number of clients who were counselled by peer educators aged 15–19 and 20–24 years.

†Refers to the number of clients who were counselled by male and female peer educators.

aRR, adjusted relative risk; Ref, reference; RR, relative risk; VMMC, voluntary medical male circumcision.

## Limitations

However, there were also limitations. First, due to deficiencies in documentation in the peer educator registers, we could not determine if the reason for non-referral (for VMMC or HTS) was related to 'need'; that is, they had already received VMMC or HTS before. We therefore could not exclude clients not in need from the analysis. In certain circumstances, like referral for contraception and STI diagnosis and treatment, we could not perform further analyses as it was not possible to establish the appropriate denominator defining the eligibility for these services. Second, we were not able to establish the exact reasons for non-referral and non-receipt of services. This requires qualitative and youth-led study approaches. Third, due to the small sample size of those referred for STI diagnosis and treatment, we lacked statistical power to carry out further analyses. Many reviews on peer education have largely focused on assessing the designs or models of peer education without focusing on the outputs and the immediate positive outcomes of peer education in relation to receipt of diagnosis and treatment of STIs, HTS, contraception and VMMC services. Therefore, there were no study results to compare with. While the results and conclusions of this study may be used in different peer education interventions, they cannot be generalised.

## Implications

The study has the following implications:
► There is a need for a review of peer education data collection tools to capture more client and peer educator data, for example, on details and quality of sessions and eligibility of clients for the various services, in line with the recently adapted revised international guidelines on CSE in Zimbabwe.[24] The

**Table 5** Factors associated with non-receipt of VMMC services among referred young people in Bulawayo and Mount Darwin, Zimbabwe (October–December 2018)

| Sociodemographic characteristics | Total referred | No receipt for VMMC services n | (%) | Unadjusted RR | (95% CI)) | Adjusted aRR | (95% CI) |
|---|---|---|---|---|---|---|---|
| Total | 1287 | 406 | (100) | | | | |
| Age (years) | | | | | | | |
| 10–14 | 657 | 202 | (30.7) | Ref | | Ref | |
| 15–19 | 433 | 136 | (31.4) | 1.02 | (0.82–1.27) | 1.24 | (0.84–1.51) |
| 20–24 | 197 | 68 | (34.5) | 1.12 | (0.85–1.48) | 1.29 | (0.91–1.83) |
| Marital status | | | | | | | |
| Single | 1275 | 403 | (31.6) | Ref | | Ref | |
| Married | 12 | 3 | (25.0) | 0.79 | (0.25–2.46) | 0.87 | (0.28–2.74) |
| School status | | | | | | | |
| In-school | 1009 | 328 | (32.5) | Ref | | Ref | |
| Out of school | 278 | 78 | (28.1) | 0.86 | (0.67–1.10) | 0.80 | (0.58–1.10) |
| Counselling session | | | | | | | |
| Individual | 221 | 102 | (46.2) | Ref | | Ref | |
| Group | 1066 | 304 | (28.5) | **0.62** | **(0.49–0.77)** | **0.52** | **(0.41–0.66)** |
| Type of setting | | | | | | | |
| Bulawayo (urban) | 748 | 284 | (38.0) | Ref | | Ref | |
| Mount Darwin (rural) | 539 | 122 | (22.6) | **0.60** | **(0.48–0.74)** | **0.52** | **(0.41–0.67)** |
| Age of peer educator (years)* | | | | | | | |
| 15–19 | 290 | 67 | (23.1) | Ref | | Ref | |
| 20–24 | 997 | 339 | (34.0) | **1.47** | **(1.13–1.91)** | 1.13 | (0.84–1.51) |
| Sex of peer educator† | | | | | | | |
| Male | 719 | 218 | (30.3) | 0.92 | (0.75–1.11) | **0.76** | **(0.62–0.94)** |
| Female | 568 | 188 | (33.1) | Ref | | Ref | |

In bold: statistically significant at p<0.05.
*Refers to the number of clients who were referred by peer educators aged 15–19 and 20–24 years.
†Refers to the number of clients who were referred by male and female peer educators.
aRR, adjusted relative risk; Ref, reference; RR, relative risk; VMMC, voluntary medical male circumcision.

review will also need to be followed with refresher training on the new concepts. This will help in determining actual output and factors associated with it.

► The project needs to consider integrating parent–child communication interventions into the VMMC–ASRH linkages project so as to mobilise parents to support young people to access services (both psycho-socially and financially). In the long run, this may also provide opportunities for home-based HTS and supervised HIV self-testing.

► While HTS are better provided through individual sessions, VMMC services are better received when group counselling sessions are provided. Thus, there is a need to focus on differentiated service provisions, depending on circumstances, even as services are integrated.

► In view of the male peer educators having higher chances of effectively referring clients for VMMC, the peer education component needs to differentiate approaches which might include pairing of female peer educators with their male counterparts to

enhance the acceptability, confidence and capacity of female peer educators to mobilise, counsel and refer for VMMC services.

► There is a need to review and possibly abolish user fees attached to the diagnosis and treatment of STIs among adolescents and young people in need of such services.

► Global standards on provision of quality health services to young people should be adopted and client satisfaction surveys should be prioritised to enhance quality service delivery.[17 25] This will help contextualise peer education into service delivery and help understand the reasons for non-referral and non-receipt of services.

## CONCLUSION

This study found varying levels of referrals ranging from 5.1% (STIs) to 58.3% (VMMC) but high levels of receipt of services among referred clients. Receipt of contraception, VMMC and HTS was high among those referred. Factors affecting non-referral included age of client, sex of

**Table 6** Factors associated with non-receipt of contraception services* among referred young people in Bulawayo and Mount Darwin, Zimbabwe (October–December 2018)

| Sociodemographic characteristics | Total referred | No receipt for Contraception services | | Unadjusted | | Adjusted | |
|---|---|---|---|---|---|---|---|
| | | n | (%) | RR | (95% CI) | aRR | (95% CI) |
| Total | 452 | 89 | (100%) | | | | |
| Age (years) | | | | | | | |
| 10–14 | 29 | 5 | (17.2) | Ref | | Ref | |
| 15–19 | 175 | 30 | (17.1) | 0.99 | (0.39–2.56) | 0.93 | (0.35–2.44) |
| 20–24 | 248 | 54 | (21.8) | 1.26 | (0.51–3.16) | 1.42 | (0.55–3.67) |
| Sex | | | | | | | |
| Male | 246 | 44 | (17.9) | 0.82 | (0.54–1.24) | 0.73 | (0.46–1.14) |
| Female | 206 | 45 | (21.8) | Ref | | Ref | |
| Marital status | | | | | | | |
| Single | 399 | 82 | (20.6) | Ref | | Ref | |
| Married | 48 | 4 | (8.3) | 0.41 | (0.15–1.11) | **0.20** | **(0.07–0.58)** |
| Divorced or separated | 5 | 3 | (60.0) | 2.92 | (0.92–9.24) | 1.38 | (0.42–4.52) |
| School status | | | | | | | |
| In-school | 116 | 30 | (25.9) | Ref | | Ref | |
| Out of school | 336 | 59 | (17.6) | 0.68 | (0.44–1.05) | 0.83 | (0.51–1.35) |
| Counselling session | | | | | | | |
| Individual | 170 | 43 | (25.3) | Ref | | Ref | |
| Group | 282 | 46 | (16.3) | **0.64** | **(0.43–0.98)** | 0.71 | (0.46–1.11) |
| Type of setting | | | | | | | |
| Bulawayo (urban) | 284 | 32 | (11.3) | Ref | | Ref | |
| Mount Darwin (rural) | 168 | 57 | (33.9) | **3.01** | **(1.95–4.64)** | **3.18** | **(1.93–5.22)** |
| Age of peer educator (years)† | | | | | | | |
| 15–19 | 49 | 18 | (36.7) | Ref | | Ref | |
| 20–24 | 403 | 71 | (17.6) | 0.48 | (0.29–0.80) | 1.00 | (0.56–1.82) |
| Sex of peer educator‡ | | | | | | | |
| Male | 180 | 26 | (14.4) | 0.62 | (0.39–0.98) | 0.81 | (0.49–1.32) |
| Female | 272 | 63 | (23.2) | Ref | | Ref | |

In bold: statistically significant at p<0.05.
*Contraceptives included condoms, injectables, oral pills and implant hormonal contraceptives.
†Refers to the number of clients who were referred by peer educators aged 15–19 and 20–24 years.
‡Refers to the number of clients who were referred by male and female peer educators.
aRR, adjusted relative risk; RR, relative risk.

peer educator and type of counselling session (individual/group), while type of setting (rural/urban), age of client and sex of peer educators affected non-receipt of services. Peer education service differentiation based on gender and service type may further enhance uptake. Advocacy efforts for user fee removal and exemptions on SRH services for young people require accelerated investment to increase service uptake. There is also a need to review the output of the peer educator project with a view to enhancing the quality of care provided to adolescents and young people. The study also recommends qualitative approaches to further understand reasons for non-referrals and non-receipt of services.

**Author affiliations**
[1]Family Health, Ministry of Health and Child Care, Harare, Zimbabwe
[2]Operational Research, International Union Against Tuberculosis and Lung Disease (The Union), Paris, France
[3]Research, National Tuberculosis, Leprosy and Lung Disease Program, Nairobi, Kenya
[4]AIDS and TB, Ministry of Health and Child Care, Harare, Zimbabwe
[5]Operational Research, International Union Against Tuberculosis and Lung Disease (The Union), Harare, Zimbabwe
[6]Operational Research, International Union Against Tuberculosis and Lung Disease (The Union), Harare, Zimbabwe
[7]Research, National Institute for Medical Research, Muhimbili Centre, Dar es Salaam, United Republic of Tanzania
[8]HIV, World Health Organization, Harare. Zimbabwe, Harare, Zimbabwe
[9]HIV, WHO, Geneva, Switzerland

[10]Research, Africaid, Harare, Zimbabwe
[11]Technical, Zimbabwe National Family Planning Council, Harare, Zimbabwe
[12]Research, All India Institute of Medical Sciences, Nagpur, India
[13]Yenepoya Medical College, Yenepoya (Deemed to be University), Mangaluru, India

**Acknowledgements** This research was conducted through the Structured Operational Research and Training Initiative (SORT IT), a global partnership led by the Special Programme for Research and Training in Tropical Diseases at the WHO. The training model is based on a course developed jointly by the International Union Against Tuberculosis and Lung Disease (The Union) and Medécins sans Frontières. The specific SORT IT programme, which resulted in this publication, was implemented by the Centre for Operational Research, The Union, Paris, France. Mentorship and the coordination/facilitation of this particular SORT IT workshop was provided through the Centre for Operational Research, The Union, Paris, France; the Department of Tuberculosis and HIV, The Union, Paris, France; the University of Washington, School of Public Health, Department of Global Health, Seattle, Washington, USA; National Institute for Medical Research, Muhimbili Centre, Dar es Salaam, Tanzania; and AIDS & TB Department, Ministry of Health & Child Care, Harare, Zimbabwe.

**Collaborators** Aveneni Mangombe.

**Contributors** AM was the principal investigator; AM, TMM, NZ, SM, SX and BM conceived the study; AM, PO, AMVK, KCT, CT, JPT, NZ, SM, AC, MS and TT designed the study protocol; AM, TMM, NZ, SM, SX, BM, JS, WA, JPT and AC read and approved the protocol; AM and TMM collected the data; AM, PO, AMVK, KCT, CT, JPT, NZ, SM, AC, MS and TT contributed to the analysis and interpretation of the data; AM, AMVK, PO, MS and TT drafted the manuscript; AM, AMVK, PO, TMM, NZ, SM, JS, WA, AC, MS, JPT, SX and BM critically revised themanuscript for intellectual content. All authors read and approved the final manuscript.

**Funding** The training course under which this study was conducted was funded by the UK's Department for International Development, The Global Fund to Fight AIDS, Tuberculosis and Malaria and the WHO. The funders had no role in study design, data collection and analysis, decision to publish or preparation of the manuscript.

**Disclaimer** The authors report no conflicts of interest.

**Competing interests** None declared.

**Patient consent for publication** Not required.

**Ethics approval** Permission to undertake the study was granted by the Ministry of Health and Child Care. Ethics approval was obtained from the Medical Research Council of Zimbabwe (MRCZ/E/223) and the Ethics Advisory Group of the International Union Against Tuberculosis and Lung Disease, Paris, France (EAG number: 62/18). As this was a retrospective analysis of deidentified routine data, the need for individual client consent was waived by both ethics committees.

**Provenance and peer review** Not commissioned; externally peer reviewed.

**Data availability statement** Data are available upon reasonable request. The corresponding author can avail the data set on request without undue reservation.

**ORCID iDs**
Aveneni Mangombe http://orcid.org/0000-0002-4057-3378
Talent M Makoni http://orcid.org/0000-0003-4303-5521
Kudakwashe C Takarinda http://orcid.org/0000-0002-2980-7735

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
