## [Reviewer comments · BMJ Open]

ARTICLE DETAILS

TITLE (PROVISIONAL)	Does peer education go beyond giving reproductive health information? A cohort study in Bulawayo and Mount Darwin, Zimbabwe
AUTHORS	Mangombe, Aveneni; Owiti, Philip; Madzima, Bernard; Xaba, Sinokuthemba; Makoni, Talent; Takarinda, Kudakwashe; Timire, Collins; Chimwaza, Anesu; Senkoro, Mbazi; Mabaya, Simbarashe; Samuelson, Julia; Ameyan, Wole; Tapera, Talent; Zwangobani, Nonhlanhla; Tripathy, Jaya; Kumar, Ajay

VERSION 1 – REVIEW

REVIEWER	Bart van den Borne Maastricht \university The Netherlands
REVIEW RETURNED	15-Oct-2019

GENERAL COMMENTS	General comments Describing, analyzing and evaluating the peer education in the context of the Zimbabwe health care provision is very relevant. In this respect the paper is interesting. The paper however, has some serious weaknesses. The paper is not very clear about the functioning and organization of the peer education. In particular the different roles of male and female peers are not clear. When and where are female peer involved and when and where are male peer involved? In the data analysis a distinction between the referral and service receipt of males and females is not made. This is relevant as there is a focus on both services (mainly) for males (voluntary circumcision) and for females (contraception). The abstract of the paper is not very clear. Also, de description of the research design is incomplete. Specific comments Abstract Research design Description of the research design is not clear. What cohort? Who are in the cohort? Mention the study concerns boys/men? Does the cohort consist of boys/males counseled by 95 different peer educators? Make that clear. it looks like the actual study units are counseled boys/males nested in 95 peer educators. Was this a cross-sectional study analyzing the correlates/determinant of referral/non-referral and correlates/determinants of referral/non-referral for particular services? Mention secondary data analysis. Participants Who were the main unit of analysis in the study? boys/males who were counseled by peer educators or peer educators themselves? That is not clear yet. Analyses seem to have been conducted on the first units so they would be the participants in your study. Peers
--

	seem to be an independent variable in the analysis of data on counseled boys/males and they were key persons in the 'intervention'. Data analysis Mention the (kind of) variables (factors) (or categories) that were included in the study. What was the dependent variable or variables in the study? referral and receipt of services? Make that clear from the start. Introduction Population of Zimbabwe in 2017 13.4 million. Is this true? Other statistics say differently. What is the relevance of knowing referral/non-referral and receipt/non-receipt of services referred for. This needs a more explicit description. Methods Research design. This is not adequate for the description of a study design. At minimum it should include mentioning the participants or population. See also comments on the description of the design in the abstract Study population. Make clear that the population and the cohort includes boys/males and girls/females. Results In intro paragraph. Not clear how group counseling was provided. Was this done in school, in the classroom? Or were groups formed after individuals indicated a need for counseling? Table 2. A distinction between boys/males and girls/females in terms of referral and service use in Table 2 would be more informative. Regarding paragraph on referral and Table 2. What was the referral pattern for girls/females? Factors associated with non-referral for HIV testing services. You talk about risk for non-referral. But what if there was no need for referral? Apparently, you were unable to make the distinction between need and no need for referral. What consequences does that have for your findings?
--	--

REVIEWER	Katrina Ortblad Univeristy of Washington
REVIEW RETURNED	21-Nov-2019

GENERAL COMMENTS	MAJOR: An overall interesting paper, that for the most part is well written. These are a number of details on the outcomes and predictors that are difficult to keep track of, thus the paper could be edited to improve clarity around these moving pieces. Additionally, the paper lacks a hypothesis/conceptual for how all these various predictors might influence referral and uptake of various services among youth. In some ways, it very much so feels like a secondary analysis of the available remaining data. Adding a framework for what the authors expect to find (based on existing literature) could add clarity and improve the strength of the paper. MINOR: Abstract
--

	 • (Design): Are you using secondary data or measuring secondary outcomes? Please clarify. • (Results): Presentation of the results is confusing. Not clear what all the outcomes being measured are – referral for what services? (maybe clarify this in the outcomes section). Switching from receipt to non-receipt of services – would be good to be consistent in directionality. Also, the denominators are not specified and the characteristics of the population (e.g., age, sex) are not described. • (Conclusions): Can you be more specific about what participants were referred for and what services they took up? These variations are not clear, and the types of services participants are being referred to vary. Introduction  • (Lines 112-121): Could move this to the methods section. • (Lines 122-123): Not clear what you mean by this. Effect of peer educators on what? Studies have been conducted where peer educators deliver interventions like HIV self-testing among female sex workers – wouldn't that be an intervention beyond distribution of information? • (Line 127): Pilot districts for what? Methods  • (Line 216): Not clear to me what a “structured proforma” is. • (Lines 220-224): Not clear to me what data sources you used to measure “referral” and “receipt” – please clarify. • (Line 230-231): Consider clearly describing your outcomes in either the section above, or a separate section. • Consider adding some conceptual framework that outlines how you hypothesize peer educator will affect non-referral of SRH services and uptake of these services. This could help center the paper, because right now there are a lot of moving pieces. Results  • (Lines 247-248): How many young people were assigned to each peer educator? • (Lines 252-262): Consider adding sample size. • (Lines 259-260): Compared to receipt of other services? Please clarify. Discussion  • (Line 370): Wouldn't your conclusions suggestion that pairing male peer educators with their male counterparts be more effective for VMMC counseling? Conclusions:  • What are the policy implications of your findings? Table/figures  • (Table 1): Could you add what percentage of peer educator-peers were M-M, M-F, F-F? • (Table 2): Consider turning this into a figure – this would make the findings pop. • (Tables 3-6): Consider presenting in 1 table that just includes aRR and n(%) for each outcome (total pop for outcome could be included above). You could always add an appendix table with the RR outcomes.
--	---

Reviewer: 1

Reviewer Name: Bart van den Borne

Institution and Country: Maastricht \university, The Netherlands

General comments: Describing, analyzing and evaluating the peer education in the context of the Zimbabwe health care provision is very relevant. In this respect the paper is interesting. The paper however, has some serious weaknesses. The paper is not very clear about the functioning and organization of the peer education. In particular the different roles of male and female peers are not clear. When and where are female peer involved and when and where are male peer involved? In the data analysis a distinction between the referral and service receipt of males and females is not made. This is relevant as there is a focus on both services (mainly) for males (voluntary circumcision) and for females (contraception). The abstract of the paper is not very clear. Also, de description of the research design is incomplete.

Response: Many thanks for acknowledging the relevance and how interesting the paper is and highlighting very important observations that have since improved the quality of the paper. The role of both male and female peer educators has been further explained in the Setting Section. The distinction between the referral and service receipt of males and females has also been made. However, the paper assesses contraception for both males and females. The research design has been further explained in both the abstract and main document to incorporate the population, intervention and setting. We have added more detail in Table 2, Line 572 - 582

Specific comments:

Abstract:

Research design:

Comment 1: Description of the research design is not clear. What cohort? Who are in the cohort? Mention the study concerns boys/men? Does the cohort consist of boys/males counseled by 95 different peer educators? Make that clear. it looks like the actual study units are counseled boys/males nested in 95 peer educators. Was this a cross-sectional study analyzing the correlates/determinant of referral/non-referral and correlates/determinants of referral/non-referral for particular services? Mention secondary data analysis.

Response: Thank you for raising this observation. We have defined the study design and have clarified the study population. It is written as "A cohort study involving all young people counselled by 95 peer educators during October-December 2018, through secondary analysis of routinely collected data". Changes made on line 43 - 44

Participants

Comment 2. Who were the main unit of analysis in the study? boys/males who were counseled by peer educators or peer educators themselves? That is not clear yet. Analyses seem to have been conducted on the first units so they would be the participants in your study. Peers seem to be an independent variable in the analysis of data on counseled boys/males and they were key persons in the 'intervention'.

Response: Many thanks for seeking clarity on this. The study population (main unit of analysis) includes young people aged 10-24 years counselled by 95 peer educators under the VMMC - ASRH linkages project in Bulawayo city and Mount Darwin district of Zimbabwe during October-December 2018. This has been clarified now. Changes made on line 46

Data analysis

Comment 3. Mention the (kind of) variables (factors) (or categories) that were included in the study. What was the dependent variable or variables in the study? referral and receipt of services? Make that clear from the start.

Response: Thank you very much for this valid comment. The abstract now clearly specifies the clients' age, sex, and marital, schooling status, type of counseling received and location and the peer educators' age and sex are the independent variables whilst "non-referral" and "non-receipt of services" are the dependent variables. Changes made on line 48 - 50

Introduction

Comment 4. Population of Zimbabwe in 2017 13.4 million. Is this true? Other statistics say differently.

Response: You are very correct that there are various sources on Zimbabwe's population. Most sources are using indirect estimates. The cited source, the 2017 Inter-Censal Demographic Survey is the most recent authentic population based survey conducted by the National Statistics Office. We suggest we retain this source, also referenced (4).

Comment 5. What is the relevance of knowing referral/non-referral and receipt/non-receipt of services referred for. This needs a more explicit description.

Response: This is a very relevant question. The paper acknowledges that one of the peer educators' roles is referring clients for service uptake. Therefore, the paper seeks to highlight the levels of referrals and service uptake among referred clients. Assessing the factors associated with these levels help fully explain factors associated with the primary outcomes of peer education.

Methods

Comment 6: Research design. This is not adequate for the description of a study design. At minimum it should include mentioning the participants or population. See also comments on the description of the design in the abstract

Response: This observation or comment is very welcome and has since been incorporated to give more detailed description of the study population. Changes made on line 127 - 131

Comment 7: Study population. Make clear that the population and the cohort includes boys/males and girls/females.

Response: Thank you for noting this. We have made the necessary additions in Line 127

Results

Comment 8. In intro paragraph. Not clear how group counseling was provided. Was this done in school, in the classroom? Or were groups formed after individuals indicated a need for counseling?

Response: Thank you for the observation. We have added more information in our peer education sub-section of the Settings section, where we feel it fits better (Line 199 – 203) We have indicated that both peer educators and young people agree on the type of counseling to be adopted, considering the sensitivity of the subject matter/topic to different age groups. The paper also indicates the different places where group counselling sessions can be conducted, including schools.

Comment 9. Table 2. A distinction between boys/males and girls/females in terms of referral and service use in Table 2 would be more informative. Regarding paragraph on referral and Table 2. What was the referral pattern for girls/females?

Response: Many thanks for this important comment in the context of being gender sensitive to our analysis. The male and female gender dynamics have been factored in on referral and receipt of services in the paragraph. Changes have been made to Table 2 in Line 571 - 581.

Comment 10. Factors associated with non-referral for HIV testing services. You talk about risk for non-referral. But what if there was no need for referral? Apparently, you were unable to make the distinction between need and no need for referral. What consequences does that have for your findings?

Response: This is a very useful comment and observation. It is true that some clients might have known their HIV status or even on ART therefore with no need for HIV testing. Our paper already acknowledges this as the first limitation of the study, due to lack of data on HIV status. However, we also know that there is low uptake of HIV testing services (HTS) by young people in the country and we have mentioned that in the introduction. Hence, we believe that the proportion of young people with a previously known HIV status will be low and hence its effect on the overall analysis will be minimal. Changes made to line 275 – 277 on holding other factors constant.

Reviewer: 2

Reviewer Name: Katrina Ortblad

Institution and Country: Univeristy of Washington

MAJOR: An overall interesting paper, that for the most part is well written. These are a number of details on the outcomes and predictors that are difficult to keep track of, thus the paper could be edited to improve clarity around these moving pieces. Additionally, the paper lacks a hypothesis/conceptual for how all these various predictors might influence referral and uptake of various services among youth. In some ways, it very much so feels like a secondary analysis of the available remaining data. Adding a framework for what the authors expect to find (based on existing literature) could add clarity and improve the strength of the paper.

Response:

Many thanks for appreciating the paper. We do not provide a priori hypothesis or conceptual framework. This was an operational research project based on routine programme data and aimed to assess how well the services are being delivered and to find out in an exploratory manner any factors associated with non-delivery/receipt of services. We thus do not see the need to include the apriori hypothesis or conceptual framework at this point. We hope this is acceptable to you.

MINOR:

Abstract

Comment 1 (Design): Are you using secondary data or measuring secondary outcomes? Please clarify.

Response: Many thanks for the question. We are using secondary data to measure primary outcomes. This is indicated specifically in the abstract on Line 43 - 44

Comment 2 (Results): Presentation of the results is confusing. Not clear what all the outcomes being measured are – referral for what services? (maybe clarify this in the outcomes section). Switching from receipt to non-receipt of services – would be good to be consistent in directionality. Also, the denominators are not specified and the characteristics of the population (e.g., age, sex) are not described.

Response: Thank you for highlighting the confusion. The paper assesses referrals and receipt of HIV testing services (HTS), contraception, diagnosis and treatment of sexually transmitted infections (STIs) and VMMC services as indicated in the abstract and main document. Switching from receipt to non-receipt of services is an intentional or deliberate effort to understand factors associated with the “gap”, being an operational study, to influence design, practice and policy. We therefore suggest to retain this.

It's true that population characteristics are very limited in the abstract – Coupled with the limited word count, the paper prioritizes key results. However, we have now added the key population characteristics in the abstract as suggested in the results section. Changes made in Line 48 – 50.

Comment 3 (Conclusions): Can you be more specific about what participants were referred for and what services they took up? These variations are not clear, and the types of services participants are being referred to vary.

Response: We welcome this question and recommendations. We have specified the type of services in the conclusion as suggested in Line 61 – 62

Introduction

Comment 4 (Lines 112-121): Could move this to the methods section.

Response: Thanks for the suggestion. We have moved majority of the paragraph to the methods section as recommended to Line 191 – 197. However, we have retained a few sentences to give a better flow of the introduction.

Comment 5 (Lines 122-123): Not clear what you mean by this. Effect of peer educators on what? Studies have been conducted where peer educators deliver interventions like HIV self-testing among female sex workers – wouldn't that be an intervention beyond distribution of information?

Response: Many thanks for seeking clarity. We intended to study the role of the peer educators in the uptake of SRH and VMMC services under the SmartLync project. We have clarified that now. Our paper already acknowledges the reviewer's observation that peer education has been found to work, especially for key populations. However, we understand that one of the key recommendations from our references 7, 8 and 9 point towards the need to contextualize peer education models. This paper is therefore not only the first ever but a unique one to assess the new peer education model in Zimbabwe.

Comment 6 (Line 127): Pilot districts for what?

Response: Many thanks for the question. The VMMC-ASRH Linkages project was piloted in Bulawayo city and Mount Darwin District. We have described it in the settings under the sub-heading VMMC-ARSH linkages project.

Methods

Comment 7 (Line 216): Not clear to me what a “structured proforma” is.

Response: Thank you for seeking clarify on this. We intended to mean a secondary data collection tool that we used to extract data from the primary registers. We have re-written it as ‘structured data collection tool’. Changes have been made in Line 225

Comment 8 (Lines 220-224): Not clear to me what data sources you used to measure “referral” and “receipt” – please clarify.

Response: Thanks for asking. Our 2nd sentence under paragraph “Data variables, sources of data and data collection” acknowledges the data source was the peer educators’ registers. Line 228-229

Comment 9 (Line 230-231): Consider clearly describing your outcomes in either the section above, or a separate section.

Response: We welcome your recommendation to clearly describe our outcomes. We have clarified the key outcome variables in the data analysis section. Line 241 - 242

Comment 10. Consider adding some conceptual framework that outlines how you hypothesize peer educator will affect non-referral of SRH services and uptake of these services. This could help center the paper, because right now there are a lot of moving pieces.

Response: Many thanks for raising this again. Like we have indicated in our response to the major comment, we do not provide an apriori hypothesis or conceptual framework as this was an operational research project based on routine programme data and aimed to assess how well the services are being delivered and to find out in an exploratory manner any factors associated with non-delivery/receipt of services. We do not see the need to include the priori hypothesis or conceptual framework at this point of the manuscript. We hope this is acceptable to you.

Results

Comment 11 (Lines 247-248): How many young people were assigned to each peer educator?

Response: Thank you for the question. Being a retrospective study, we did not assign young people to any peer educator, neither is it part of the project’s design also.

Comment 12 (Lines 252-262): Consider adding sample size.

Response: Thank you for the recommendation. As an operational study, our paper highlights the inclusion of all the eligible study participants in the study area/period. Additionally we have also factored in gender disaggregated data where possible.

Comment 13 (Lines 259-260): Compared to receipt of other services? Please clarify.

Response: Thanks for the question. However, not sure if the reviewer is really commenting on lines 259-260 or wanted 269 – 271. If so, this section is referring to factors associated with non-referral for VMMC services. The paper is comparing strength of associations on factors associated with non-referral for VMMC expressed using adjusted relative risks (aRR).

Discussion

Comment 14 (Line 370): Wouldn't your conclusions suggestion that pairing male peer educators with their male counterparts be more effective for VMMC counseling?

Response: Thank you this recommendation. However, we assume the reviewer want us to qualify the need for pairing of female peer educators with male counterparts in the context of VMMC counseling and referral. If so, we have factored in this comment under Implications subsection. Changes have been made in line 386 - 390

Conclusions:

Comment 15. What are the policy implications of your findings?

Response: This question is most welcome. We had missed a recommendation on policy influence. We have since made a call for accelerated investment towards user fees removal and exemptions for young people to increase service uptake on Line 391 – 392 and 405 - 406

Table/figures

Comment 16. (Table 1): Could you add what percentage of peer educator-peers were M-M, M-F, F-F?

Response: Thank you for this comment. The project doesn't capture this information. However, we have provided the gender disaggregation of peer educators by district on Line 541 - 569

Comment 17. (Table 2): Consider turning this into a figure – this would make the findings pop.

Response: We appreciate that a figure can help visualize the data better. However, given an earlier recommendation to gender disaggregate our data as much as possible, we have since added more information. We have tried different figure formats and they are coming out too cluttered. We therefore recommend we maintain the improved/new table format.

Comment 18. (Tables 3-6): Consider presenting in 1 table that just includes aRR and n(%) for each outcome (total pop for outcome could be included above). You could always add an appendix table with the RR outcomes.

Response: Thank you for the comment. However, may you please retain the current format as it provides all the details in one table – numbers, percentages and both unadjusted and adjusted RR and 95% CI

VERSION 2 – REVIEW

REVIEWER	Bart van den Borne Maastricht University, the Netherlands
REVIEW RETURNED	11-Jan-2020

GENERAL COMMENTS	The authors have adequately addressed my previous comments on the first draft of this paper.
--

REVIEWER	Katrina Ortblad University of Washington
REVIEW RETURNED	24-Jan-2020

GENERAL COMMENTS	I thank the authors for addressing all my comments. The paper has improved.
---